# Insights into COVID-19 Vaccine Development Based on Immunogenic Structural Proteins of SARS-CoV-2, Host Immune Responses, and Herd Immunity

**DOI:** 10.3390/cells10112949

**Published:** 2021-10-29

**Authors:** Jitendra Kumar Chaudhary, Rohitash Yadav, Pankaj Kumar Chaudhary, Anurag Maurya, Nimita Kant, Osamah Al Rugaie, Hoineiting Rebecca Haokip, Deepika Yadav, Rakesh Roshan, Ramasare Prasad, Apurva Chatrath, Dharmendra Singh, Neeraj Jain, Puneet Dhamija

**Affiliations:** 1Department of Zoology, Shivaji College, University of Delhi, New Delhi 110027, India; jnujitendra@gmail.com (J.K.C.); nimita@shivaji.du.ac.in (N.K.); dy_277@yahoo.com (D.Y.); rakesh.biotech85@gmail.com (R.R.); 2Department of Pharmacology, All India Institute of Medical Sciences (AIIMS), Rishikesh 249201, India; 3Molecular Biology & Proteomics Laboratory, Department of Biotechnology, Indian Institute of Technology (IIT), Roorkee 247667, India; pkchaudharydu@gmail.com (P.K.C.); ramasare.prasad@bt.iitr.ac.in (R.P.); apurvachatrath@gmail.com (A.C.); zooldm85@gmail.com (D.S.); 4Department of Botany, Shivaji College, University of Delhi, New Delhi 110027, India; anuragvns.maurya@gmail.com; 5Department of Basic Medical Sciences, Unaizah College of Medicine and Medical Sciences, Qassim Uniersity, Unaizah 51911, Saudi Arabia; o.alrugaie@qu.edu.sa; 6Medical Allied ICU, All India Institute of Medical Sciences, Rishikesh 249201, India; rebeccahaokip9620@gmail.com; 7Department of Cancer Biology, CSIR-Central Drug Research Institute, Lucknow 226031, India; neeraj.monc@aiimsrishikesh.edu.in

**Keywords:** SARS-CoV-2, coronavirus disease 19, vaccination, infectious disease, pandemic, immune response, herd immunity

## Abstract

The first quarter of the 21st century has remarkably been characterized by a multitude of challenges confronting human society as a whole in terms of several outbreaks of infectious viral diseases, such as the 2003 severe acute respiratory syndrome (SARS), China; the 2009 influenza H1N1, Mexico; the 2012 Middle East respiratory syndrome (MERS), Saudi Arabia; and the ongoing coronavirus disease 19 (COVID-19), China. COVID-19, caused by SARS-CoV-2, reportedly broke out in December 2019, Wuhan, the capital of China’s Hubei province, and continues unabated, leading to considerable devastation and death worldwide. The most common target organ of SARS-CoV-2 is the lungs, especially the bronchial and alveolar epithelial cells, culminating in acute respiratory distress syndrome (ARDS) in severe patients. Nevertheless, other tissues and organs are also known to be critically affected following infection, thereby complicating the overall aetiology and prognosis. Excluding H1N1, the SARS-CoV (also referred as SARS-CoV-1), MERS, and SARS-CoV-2 are collectively referred to as coronaviruses, and taxonomically placed under the realm *Riboviria,* order *Nidovirales,* suborder *Cornidovirineae,* family *Coronaviridae,* subfamily *Orthocoronavirinae*, genus *Betacoronavirus*, and subgenus *Sarbecovirus*. As of 23 September 2021, the ongoing SARS-CoV-2 pandemic has globally resulted in around 229 million and 4.7 million reported infections and deaths, respectively, apart from causing huge psychosomatic debilitation, academic loss, and deep economic recession. Such an unprecedented pandemic has compelled researchers, especially epidemiologists and immunologists, to search for SARS-CoV-2-associated potential immunogenic molecules to develop a vaccine as an immediate prophylactic measure. Amongst multiple structural and non-structural proteins, the homotrimeric spike (S) glycoprotein has been empirically found as the most suitable candidate for vaccine development owing to its immense immunogenic potential, which makes it capable of eliciting both humoral and cell-mediated immune responses. As a consequence, it has become possible to design appropriate, safe, and effective vaccines, apart from related therapeutic agents, to reduce both morbidity and mortality. As of 23 September 2021, four vaccines, namely, Comirnaty, COVID-19 vaccine Janssen, Spikevax, and Vaxzevria, have received the European Medicines Agency’s (EMA) approval, and around thirty are under the phase three clinical trial with emergency authorization by the vaccine-developing country-specific National Regulatory Authority (NRA). In addition, 100–150 vaccines are under various phases of pre-clinical and clinical trials. The mainstay of global vaccination is to introduce herd immunity, which would protect the majority of the population, including immunocompromised individuals, from infection and disease. Here, we primarily discuss category-wise vaccine development, their respective advantages and disadvantages, associated efficiency and potential safety aspects, antigenicity of SARS-CoV-2 structural proteins and immune responses to them along with the emergence of SARS-CoV-2 VOC, and the urgent need of achieving herd immunity to contain the pandemic.

## 1. Introduction

The positive, single-stranded RNA genome-containing SARS-CoV-2 is the causative virus of coronavirus disease 19 (COVID-19), which belongs to the species severe acute respiratory syndrome-related coronavirus and the genus *Betacoronavirus*. It is taxonomically clustered with the 2002–2003 outbreak-causing SARS-CoV, also referred to as SARS-CoV-1 [1,2]. Coronaviruses are known to infect both humans and a range of animal species, as shown using animal models for COVID-19, as well as by the detection of presence of natural infection in ferrets, domestic cats, Golden Syrian hamsters, cynomolgus, rhesus macaques, rabbits, dogs, lions, minks, and tigers [3,4], with a wide spectrum of pathological symptoms, severity, and aetiological consequences. In humans, COVID-19 is typically characterized by the development of acute respiratory distress syndrome (ARDS) in severe patients as a consequence of excessive inflammatory cytokine-led irreversible histological destruction of the bronchial and alveolar epithelium, with pathological features similar to ones observed during the SARS-CoV pandemic in 2002–2003. However, mild to moderately ill patients show a spectrum of symptoms, such as fever, cough, tiredness, sore throat, chills, runny nose, headache, and chest pain amongst others, with some overlapping symptoms, while others may be individual patient-specific. SARS-CoV-2 has caused nearly 229 million lab-confirmed infections and 4.7 million reported deaths worldwide as of 23 September 2021, with an average 2% death rate so far. Responding to such a horrifying natural phenomenon, a global vaccination programme has been launched to break the chain of spread and effectively contain the infection. As of 22 September 2021, there has been an administration of at least one dose of the COVID-19 vaccine to 44% of individuals, whereas 32% of individuals stand fully vaccinated after the receipt of both doses globally (WHO Coronavirus (COVID-19) Dashboard; WHO Coronavirus (COVID-19) Dashboard With Vaccination Data).

Retrospectively, the initial outbreak of COVID-19 was traced back to 8 December 2019, whereas the Wuhan Municipal Health Commission informed the World Health Organization (WHO) and common people of the occurrence of pneumonia of an unknown underlying cause on 31 December 2019 [5]. Furthermore, metagenomic RNA sequencing and a bronchoalveolar lavage (BAL) fluid sample analysis by several independent groups of Chinese researchers identified the causative agent of COVID-19 as SARS-CoV-2 (initially called 2019-nCoV), which had never been recorded in human history. Amongst the initial 27 hospitalized patients at the outset of the outbreak, most cases were found to be associated with the Huanan Seafood Wholesale Market, indicative of animal-to-human transmission (zoonosis). Within a short period of time, a few more individuals contracted this disease despite no direct exposure to the abovementioned seafood market, indicating human-to-human transmission (anthroponosis), as well as the occurrence of nosocomial infection in health-care facilities [6,7]. SARS-CoV-2 quickly spread far and wide, covering all 34 provinces of China, as well as other nations, islands, and territories located over seven continents with varying numbers of daily reported cases. Considering the emerging condition, the WHO declared this viral outbreak as a public health emergency of international concern (PHEIC) on 30 January 2020 [8]. By February 2020, SARS-CoV-2-induced COVID-19 reached an epidemic proportion, with reports of around 3000 laboratory-confirmed infections per day. Assessing the seriousness of the situation, China swung into action forthwith, implementing unprecedented public health measures, such as restriction on outdoor activities, social congregation, travel, and transportation in the city of Wuhan and the countryside. Such a series of measures and their strict ground-level implementation substantially helped contain the infection, with an incremental decline in daily lab-confirmed cases through time. As a result of such countermeasures, there has been a drastic downward slide in daily reported cases in China with time, placing it currently at the 110th position (in terms of cumulative infection) from the initial 1st position, among the 222 nations/territories in the world (https://www.worldometers.info/coronavirus/) (accessed on 17 October 2021).

In contrast to China, SARS-CoV-2 started spreading across the globe with ever increasing number of daily lab-confirmed cases, leading to huge disruptions to normal lives and livelihoods, as well as proportionate psychosomatic debilitation and death [9]. Currently, the world is going through an existential crisis with an average 6–8 lakhs of daily reported cases and thousands of deaths, and the three most affected countries, in terms of cumulative cases, are the USA, India, and Brazil. The continuance of the current pandemic must be attributed to several variables, such as a high transmissibility and an intrinsically inbuilt immune evasiveness of the virus, the presence of a huge proportion of asymptomatic carriers among infected individuals, a global emergence of variants of concern (VOC), inappropriate COVID-19 behaviour owing to the prevalence of callousness and medical/health illiteracy in the general population, virus-induced cytokine storm, a lack of specific anti-SARS-CoV-2 medicines, and standard treatment modality [10,11,12,13,14]. This has compelled researchers and clinicians to hurriedly repurpose the FDA-approved drugs intended against non-related infections [15,16,17,18,19], develop novel therapeutic molecules/drugs, cell-based therapy, convalescent plasma therapy, antibody cocktails, and vaccines to combat the current pandemic as many researchers have reviewed previously [20,21,22,23,24]. Currently, the most common consensus amongst the various stakeholders, including doctors and researchers regarding the immediate way out of this pandemic, is the vaccine and successful implementation of the vaccination programme to achieve herd immunity. As per the well-established herd immunity/population immunity theory, the vaccination of a population up to the herd immunity threshold (HIT) may contain and combat the SARS-CoV-2 spread, as well as provide indirect protection to the susceptible and immunocompromised individuals [25]. Therefore, we discuss various immunogenic epitopes on SARS-CoV-2 structural (S, M, E, and N) and non-structural proteins (various ORF-encoded proteins), the development of vaccines based on such epitopes, innate and adaptive immune responses, as well as vaccines’ nature, efficacy, doses along with gap and the routes of administration, and known side effects and limitations, especially with regard to SARS-CoV-2 VOC.

## 2. Nature of Vaccines

The central dogmatic theme of the multitude of vaccines—irrespective of the nature, procedure, and platform of manufacturing—primarily revolves around providing protection against pathogens, such as bacteria, virus, fungi, and endoparasites. Conventional vaccine strategies, involving inactivated and live-attenuated pathogens, have provided durable and effective protection against a variety of dreaded infectious diseases, but may not be useful for non-infectious diseases, such as cancer. In addition, they require significantly longer duration for customization, development, and manufacturing. In contrast, modern vaccine strategies utilized for mRNA and DNA vaccines potentially meet the need for a rapid development and industry-scale deployment prerequisite for constantly emerging and evolving RNA viruses, including SARS-CoV-2 [26]. Immune protection is accomplished by the activation of both humoral and cell-mediated responses post vaccination and natural infection. Although the protective immune responses following vaccination are quite similar to the ones observed post natural infection, the former acts as a prophylactic measure, and has helped in the eradication (e.g., small pox) and drastic reduction (e.g., polio and measles) in a myriad of infectious diseases worldwide. The protective immune responses in terms of CoP (correlates of protection) following vaccination may differ depending on the individual, age, sex, immune status, race, demography, genetics/epigenetics and environmental factor, ethnicity, etc. Various types of vaccines against COVID-19 are being developed, authorized, and then administered globally (Table 1; Figure 1 and Figure 2). A category-wise description of conventional and modern vaccines, strategies, potential safety, and risks are discussed below.

### 2.1. Inactivated or Killed Virus (SARS-CoV-2) Vaccine

Such vaccines involve a thermally or chemically inactivated/killed viruses following their propagation in Vero cells—a cell line derived from kidney epithelial cells of the African green monkey—and most commonly used for virus cultures [27]. Chemical inactivation, based on formaldehyde or multiple alkylating agents, is preferred over the thermal one, as the former helps in maintaining and preserving the integrity of the panoply of the immunogenic epitopes of viruses in a better way [28]. Methodology pertaining to the development of an inactivated vaccine is well established and, hence, can be rapidly scaled up to meet the urgent need for the control of an epidemic/pandemic [27]. The important advantages include a greater safety and stability even without refrigeration; however, they generate weaker immune responses and, therefore, often require the administration of booster doses [29,30]. This category of vaccines has been used to evoke a protective immune response against a multitude of pathogenic viruses over several decades [31]. Examples of authorized SARS-CoV-2 vaccines under this category are CoronaVac, BBIBP-CorV, Covaxin (BBV152), WIBP-CorV, CoviVac, and QazVac (QazCovid-in) (Table 1).

### 2.2. Live-Attenuated Virus (SARS-CoV-2) Vaccine

Such a vaccine consists of attenuated or crippled live viruses with least pathogenicity, on the one hand, but an intact immunogenicity and transient growth potential, on the other [50]. Attenuated viruses show the highest immunogenicity, and closely mimic natural infection without causing the actual disease [51]. The process of attenuation is often carried out by prolonged in vitro or in vivo passages under abnormal conditions, as well as reverse genetic mutagenesis [52]. Moreover, recent technological advancements have made it possible to selectively delete virulence gene(s), thereby crippling virus pathogenic features responsible for causing disease. Historically, the most successful vaccines under this category are the Oral Polio Vaccine (OPV), bacillus Calmette–Guérin (BCG) vaccine, and measles vaccine [53,54]. The main advantages include strong immunogenic responses and the likelihood of life-long protection; however, they require refrigerated storage, and an attenuated virus may seldom mutate to a virulent form [55]. Apart from being effective against tuberculosis, the BCG vaccine is well known to confer anti-viral immunity by inducing proinflammatory cytokine production (IL-6, TNF-α, IFN-γ, and IL-1β), and also evokes ‘trained immunity’, wherein CD4 and CD8 memory cells are activated in an antigen-independent manner. This leads to the generation of a non-specific heterologous protective immune response against non-related infectious diseases, such as neonatal sepsis and respiratory infections, and thereby reduces mortality amongst at-birth BCG-vaccinated children [56,57,58]. Moreover, the BCG vaccination of murine experimental models has shown a protective effect against non-mycobacterial infections [59]. In light of such empirical evidence, the Bacillus Calmette-Guerin (BCG) vaccine is being studied through phase one and two/three clinical trials (ClinicalTrials.gov Identifier: NCT04327206 and NCT04328441) (Table 1) for its protective effect, if any, against SARS-CoV-2 as well. Berg MK et al. carried out a study, comparing countries with mandated BCG vaccination with that of non-mandated ones until at least the year 2000 and, after adjusting for median age, gross domestic product (GDP) per capita, population size, population density, and the net migration rate, they found a significantly reduced growth rate of COVID-19 cases in BCG-mandated countries. This suggests that a BCG vaccination of sufficient proportion of the population up to the herd immunity threshold (HIT) may be quite effective in the fight against COVID-19 [35].

### 2.3. Virus-Like Particle (VLP) Vaccine

Such a vaccine involves non-replicating (as they lack a genome) and non-infectious particles, possessing virus immunogenic structural proteins. In this strategy, viral structural proteins are expressed and self-assembled in vitro, forming a particulate morphology, which structurally and immunogenicity-wise resembles the real virion [60]. Some of the examples of SARS-CoV-2 vaccines under development in this category are NVX-CoV2373 and EuCorVac-19 (nanoparticle vaccine) (Table 1). Nonetheless, lots of aspects of such a vaccine category/type remain unexplored and, hence, requires more empirical work.

### 2.4. Peptide or Protein Subunit Vaccine

Such a vaccine involves key immunogenic peptides or intact viral proteins; for instance, the spike (S) protein in the case of SARS-CoV-2, which is capable of inducing a protective immune response. These immunogenic proteins are expressed in vitro in a range of biological hosts, including bacteria, yeast, insects, and mammals, as well as chemically synthesized. Protein subunit vaccines can be rapidly developed, upgraded, and upscaled for mass production to meet the ever increasing global demand. Further, they have a low reliance on refrigerated storage [61], and also safety issues concerning the whole virus vaccine (killed and attenuated vaccines), such as virulence reversal, incomplete inactivation and attenuation, and pre-existing immunity, can be completely ruled out. Moreover, such a vaccine can be precisely designed to contain well-defined immunogenic epitope(s) capable of inducing a potent neutralizing antibody formation and activation of T cells to enhance the overall vaccine effectiveness. An example of the approved/authorized SARS-CoV-2 vaccine under this category is EpiVacCorona (Table 1).

### 2.5. Nucleic Acid (DNA and mRNA) Vaccine

Nucleic acid (DNA and RNA) vaccines are considered as a modern, cost-effective, promising, and easy-to-design and develop vaccine candidates, with huge therapeutic potential [62,63]. They are produced rapidly, and can be upgraded and scaled up to match the dynamic requirement as there is no need for growing viruses or viral proteins inside live cells in tissue culture laboratory. For a DNA vaccine, generally recombinant plasmid containing immunogenic antigen-encoding gene(s) is administered intramuscularly, wherein the antigen is produced through a sequential transcription-to-translational molecular process, triggering protective humoral and cellular immune responses [49,64]. Such a protective efficacy has been shown in both experimental set-ups—in vitro and in vivo—, leading to the approval of a few DNA vaccines for veterinary applications in the past [65]. Advancements in genetic engineering over the past decades, including species-specific codon optimization, novel formulation, and a more effective delivery system, have resulted in an increased DNA vaccine-encoded antigen production on a per-cell basis, eliciting enhanced humoral and cellular responses [66]. Although a DNA vaccine offers several advantages over conventional vaccines, nevertheless, there are some potential safety concerns as well. These concerns entail the likelihood of the genomic integration of the antigen-encoding recombinant plasmid DNA molecule, autoimmunity development, and acquisition of antibiotic resistance [66].

The genomic integration of foreign DNA, including the plasmid, may cause the inactivation of tumour suppressor genes and the hyper activation of proto-oncogenes, collectively referred to as insertional mutagenesis. Moreover, there could be an induction of chromosomal breaks or rearrangements and, hence, chromosomal instability. So far, evidences suggesting genomic integration are scarce and, if any, the rate is far below the normal (three orders of magnitude) compared with the spontaneous mutation frequency shown with regard to four different DNA vaccine plasmids in non-human mammals [67]. Similarly, there are no convincing evidences of the induction of autoimmunity by DNA vaccines in human and non-human primates during preclinical and preliminary clinical studies. So far as the acquisition of antibiotic resistance following plasmid DNA-based vaccination is concerned, no-antibiotic-selection strategy is being developed to preclude any such likelihood [68,69]. Nevertheless, current plasmid DNA vaccines often rely on a kanamycin-based selection marker, which is not generally used to treat bacterial infection in human. Considering such pros and cons, a DNA vaccine for SARS-CoV-2 is being developed and tested in preclinical and clinical studies [70]. ZyCoV-D, developed by Ahmedabad (headquarter)-based Zydus Cadila, and approved by DCGI, India, is the first plasmid DNA vaccine for SARS-CoV-2 in humans, and is currently undergoing clinical trial (CTRI/2020/07/026352)./).

The mRNA vaccine consists of in vitro synthesized/transcribed (IVT) functional mRNA, containing flanking UTRs, a 5′ cap, and a poly(A) tail, which potentially encodes the viral antigen via direct translation in host cell cytoplasm following administration [71]. There are two categories of mRNA vaccines: non-replicating mRNA and virally derived self-replicating mRNA. Whereas conventional mRNA-based vaccine is capable of encoding the antigen of interest, the self-replicating mRNA-based vaccine encodes both the antigen, as well as the viral replication machinery for intracellular mRNA amplification [72]. Relatively, mRNA vaccines are inherently less stable compared to DNA vaccines, necessitating better care at the level of handling and storage [73]. In general, in vitro synthesized mRNA molecules are allowed to complex with a lipid nanoparticle carrier, and then are delivered in vivo, wherein they undergo translation in a manner similar to any functional cellular mRNA, owing to their structural resemblance. The antigenic protein, so synthesized, undergoes requisite post translational modifications. The lack of the requirement of toxic chemicals or a cell culture for manufacturing an mRNA vaccine helps to avoid risks linked to conventional vaccine platforms as explained earlier. Moreover, the theoretical concern regarding the genomic integration of a vector does not exist in the case of the mRNA vaccine. Examples of approved SARS-CoV-2 mRNA vaccines under this category are Comirnaty (BNT162b2) and Moderna COVID-19 Vaccine (mRNA-1273). Lastly, these vaccines have been proven to be among the most efficient authorized vaccines for COVID-19, and are capable of eliciting both humoral and cellular immune responses [66] (Table 1; Figure 2A).

### 2.6. Virus-Vectored Vaccine

Such vaccines involve a non-replicating (replication-deficient) or weakly replicating viral vector backbone bioengineered to carry viral antigen-encoding gene(s). This platform has an investigation-based well-established track record and, owing to some important characteristics, such as safety, easily achievable genetic malleability, strong potential to induce cell-mediated immunity (T cell), and the lack of an adjuvant requirement, they are quite convenient to design and develop [74,75]. Further, such a recombinant viral-vectored vaccine does not require a booster dose administration, and also shows good respiratory mucosal tropism. Some of the best studied viral vectors are adenovirus, measles, and vaccinia [74]. Viral antigens are produced in the host following vaccination, triggering protective immune response. Some of the examples of authorized SARS-CoV-2 vaccines under this category are COVID-19 Vaccine AstraZeneca (AZD1222; also known as Vaxzevria and Covishield), Sputnik V, Sputnik light, and Convidicea (Ad5-nCoV) (Table 1; Figure 2).

## 3. Immune Responses to SARS-CoV-2 Structural and Non-Structural Proteins

There are four SARS-CoV-2 structural proteins; namely, Spike (S), Membrane (M), Envelope (E), and Nucleocapsid (N). The former three remain differentially embedded in the viral lipid bilayer envelope derived from the host cell membrane, and later remains complexed with the viral +ss mRNA (positive, single-stranded mRNA) genome (Figure 3A). The lengths of these structural proteins (amino acids) are mentioned in Table 2. Each structural protein contains multiple domains with specific biological functions (Figure 3B–E). Furthermore, the antigenic propensity score of each structural protein has been predicted individually using a bioinformatics tool (Table 2) (http://imed.med.ucm.es/Tools/antigenic.pl) (accessed on 17 October 2021). As per the empirically established norm, protein with more than a 0.8 antigenic propensity score is generally considered highly suitable for vaccine development [61,76].

### 3.1. Activation of Innate Immune Response against SARS-CoV-2

Following virus entry into body, a cellular fight ensues immediately between the virus and innate immune cells—the first line of cellular defence—, and the outcome may either be the rapid clearance of the pathogenic virus or the establishment of infection owing the virus’s immune evasion capability, resulting in the occurrence of the disease [77,78,79]. The innate immune response against SARS-CoV-2 is accomplished by the functional activation of a multitude of lung-resident myeloid cells, such as alveolar macrophages (AM), neutrophils, monocytes, and dendritic cells, despite the viral immune evasiveness. Moreover, non-lung resident innate immune cells, such as eosinophils, are also known to respond against viruses. For example, eosinophils may act against a respiratory syncytial virus and influenza [80] through antiviral activity and also serve as antigen-presenting cells (APCs), thereby facilitating an adaptive immune response by lymphoid cells. In fact, several studies on COVID-19 hospitalized patients [80,81,82], reviewed by Azkur et al. [83], showed a decline in eosinophil counts below the normal value (eosinopenia) in a significant number of patients; therefore, it may be used as a viable COVID-19 diagnostic marker and/or predictor of disease severity. These innate immune cells are equipped with multiple receptors capable of sensing diverse viral components, and respond by secreting cytokines and interferons (IFNs) in order to neutralize invading viruses [84]. For instance, the presence of pattern recognition receptors (PRRs) on a cell membrane and in the cytosol of innate immune cells helps them sense and recognize foreign factors with characteristic molecular patterns, such as pathogen-associated molecular patterns (PAMPs), as well as damage and/or infection-led endogenous molecules with typical patterns, called damage-associated molecular patterns (DAMPs) [85]. Most common PRRs include Toll-like receptors (TLRs) and RIG-I-like receptors, whose RNA virus (SARS-CoV, SARS-CoV-2, and MERS) identification capability and the subsequent induction of a type I IFN response and IFN-stimulated gene (ISG) expression potential are well known [86]. Owing to the viral neutralization capability, an innate immune system is often modulated by pathogenic organisms for immune evasion and survival. Following activation, the innate immune system evokes protective or pathogenic inflammatory responses (cytokine storm) depending on the category, source, structure, abundance, and duration of stimuli [87].

The IFN-induced JAK–STAT (Janus kinases–signal transducer and activator of transcription proteins) pathway activates several downstream IFN-stimulated genes (ISGs), triggering antiviral activities of host cells, and thereby neutralizes viruses [84]. COVID-19 patients on the lower side of the disease severity spectrum (mild to moderate) showed a higher IFN response that considerably declined in critical patients [88]. This was further supported by studies, underscoring the importance of IFN signalling as an inbuilt mechanism of cellular defence against SARS-CoV-2. Therefore, any perturbation in this response, in terms of a loss of function in responsive genetic loci, regulating Toll-like receptor-3 (TLR3) and IFN-mediated immunity, autoantibodies generation against type I IFN-α2, and IFN-ω, may lead to a life-threatening COVID-19 condition as observed in thousands of patients worldwide [89,90,91]. Wang et al. nicely compiled the crucial role of autoantibodies (AAb) against excessively produced cytokines, growth factors, chemokines, complement proteins, and even cell surface proteins, thereby compounding the adverse effect of the pathological activation of the innate immune system, disease progression, and clinical outcome [91]. In fact, the level of these autoantibodies against immune-related proteins (part of exoproteome) is positively correlated with COVID-19 severity, implicating their direct involvement in the immunopathology of the disease. Fortunately, these autoantibodies were absent in healthy individuals, as well as infected individuals with no (asymptomatic) or milder symptoms [89]. This suggests the need for expanding the current therapeutic regimen for COVID-19, involving the attenuation of autoantibodies to achieve a better clinical outcome. A consideration of the abovementioned facts may lead to the identification of COVID-19-susceptible and/or genetically predisposed individuals in advance, allowing clinicians to save their lives by employing available prophylactic measures, such as vaccines and effective antibody cocktails. An examination of hundreds of moderate to severe COVID-19 patients revealed a consistently elevated level of IFNs and viral load throughout the course of the disease. This suggests a substantial correlation between the levels of IFNs and/or TNF-α (tumour necrosis factor-α) and virus quantity, indicating a likelihood of viral load-induced production of cytokines [92].

Recent studies looking at the underlying mechanism of the SARS-CoV-2 viral RNA sensing pathway led to the identification of canonical intracellular sensor/receptor molecules, such as melanoma differentiation-associated protein 5 (MDA5), the laboratory of genetics and physiology 2 (LGP2), and nucleotide-binding oligomerization domain-1 (NOD1), in lung epithelial cells. Upon recognition, the IFN response is modulated by a set of crucial transcription factors, such as IRF3, IRF5, and NF-κB/p65, during the course of COVID-19 [93]. The activation of MyD88 and TRIF-mediated NF-κB and IRF3 signalling leads to the production of proinflammatory cytokines, including the tumour necrosis factor (TNF), interleukin (IL)-6, IL-1β, and type I/III IFNs. Thereafter, I/III IFNs, including IFNα, IFNβ, IFN_ω_, and IFNλ, cause the induction of an antiviral programme aimed at neutralizing viruses [94,95]. The in vitro study on the primary human airway epithelia (HAE) and cell lines showed a strong induction of type I and III interferons (IFNs) following SARS-CoV-2 infection. Often, an exogenous IFN exposure leads to a strong inhibition of viral replication; however, the scenario in lung cells is polar opposite, as even a substantially high level of IFN production in response to SARS-CoV-2 infection fails to check the same [96]. This requires further study regarding the relationship between SARS-CoV-2 replication, and the crucial timing and intensity of the IFN response, which may reveal the underlying reason of COVID-19 severity and manifestation.

SARS-CoV-2 has multipronged immune evasion strategies, including the inhibition of the IFN response. Cumulative evidences suggest that the inhibition of type I IFN occurs at multiple levels by SARS-CoV-2 structural and non-structural proteins following cellular invasion, thereby effectively thwarting interferon-mediated innate and adaptive anti-viral responses [97,98,99]. For instance, NSP1 binds to 18S rRNA, as well as induces cleavage in 5′-capped host mRNAs, thereby causing a reduction and/or global suppression in mRNA translation. In addition, it also blocks the post transcriptional mRNA export from the nucleus to cytosol by interfering with the working of the nuclear pore complex (NPC) protein 93 (Nup93) [100]. Furthermore, promoter activity pertaining to IFN-stimulated response elements (ISREs) has also been observed to be supressed by SARS-CoV-2 NSP1 [101]. Similarly, NSP3 may interact with IRF3 protein, preventing its dimerization, phosphorylation, and nuclear translocation [102], apart from inhibiting the NF-κB signalling pathway [103], thereby thwarting the antiviral response. An endoplasmic reticulum (ER) is induced to form a double-membrane vesicle (DMV) by the collective action of NSP3, NSP4, and NSP6. DMV encapsulates the PAMP-bearing viral replicase complex, thereby disabling the sensing of viral RNA in cytosol [104]. Furthermore, several host proteins involved in the antiviral response, such as mRNA-decapping protein 1a (DCP1a), NF-κB essential modulator (NEMO) 80, and STAT2, are enzymatically processed by SARS-CoV-2 NSP5 [105,106]. Both NSP8 and NSP9 inhibit the vesicular trafficking of protein to the cell membrane. A molecular interaction mapping study put forth the interaction of NSP13, NSP15, and ORF9b with TBK1, NRDP1, and TOMM70 (outer mitochondrial membrane receptor), respectively. Similarly, ORF6 may show an interaction with translocator proteins, such as the IFN-induced NUP98–RAE1 nuclear export complex and KPNA2, thereby affecting the translocation of key transcription factors, including IRF3, IRF7, and STAT1, as well as the inhibition of STAT1 and STAT2 phosphorylation, leading to the considerable suppression of IFN signalling and, hence, antiviral response [101,107,108]. Considering the abovementioned facts, researchers need to strategize and adopt a multipronged therapeutic regimen to contain the virus.

### 3.2. Activation of Adaptive (Humoral and Cell-Mediated) Immune Response against SARS-CoV-2

The spike (S) glycoprotein is a homotrimeric molecule, which forms as a result of the molecular association of three identical polypeptides, each containing 1273 amino acid residues. Each polypeptide chain/monomeric unit of S (spike) protein possesses two distinct subunits, namely, S1 (membrane-distal subunit) and S2 (membrane-proximal subunit), owing to the presence of a likely cleavage site for TMPRSS2 protease [109]. Together, they show multiple distinct, orderly arranged functional domains with their respective functions. For instance, the receptor-binding domain (RBD), consisting of around 200 amino acid residues, is located within the S1 subunit (often abbreviated as RBD/S1), whose molecular interaction (via the receptor-binding motif) with the ACE2 (angiotensin-converting enzyme 2) receptor is facilitated by down-to-up conformational transitioning. The S2 subunit with its multiple distinct domains, such as the fusion peptide (FP), central helix (CH), connecting domain (CD), and two heptad repeat (HR1/2) domains, helps mediate the fusion of viral envelope with that of the host cell membrane, eventually allowing the cellular entry of a virus (Figure 3B) [110,111]. Generally, each SARS-CoV-2 particle (virion) bears around 50–100 homotrimeric S glycoprotein assemblages that account for its extensive crown-like appearance. Documented evidence has clearly demonstrated occurrence of a subtle structural rearrangement/conformational change in SARS-CoV-2 S protein following the recognition of ACE2 on the host cell [112], facilitating the cellular entry of a virus.

In general, the SARS-CoV-2 S protein possesses considerable antigenic propensity (1.014) as predicted and tabulated (Table 2) [61], indicative of its capability of evoking an immune response, resulting in the production of neutralizing antibodies (nAbs), as well as domain-specific antibodies, which have been fully substantiated by an in vivo study [113]. Therefore, consideration of such a relevant aspect while designing a vaccine has proven to be quite beneficial. In general, the induction of the virus-specific IgM class of antibody is followed by the IgG antibody class (class switching) post SARS-CoV-2 infection, often within 1–2 weeks after symptom onset [114]. However, owing to class switching, the level of IgG supersedes over IgM. Therefore, the ratio of virus-specific IgM to IgG must decline, or the level of IgG goes up with the increase in the duration post infection, which is used for diagnostic purposes. Surprisingly, S protein-specific IgA (anti-S IgA) has been found to peak even before IgM in certain patients. Nevertheless, the reason is not yet understood, and awaits further investigation into the implication of the IgA class of antibody, especially in mucosal immunity owing to its secretory nature [115]. In fact, an IgA titer, compared with both IgG and IgM, showed a lesser decline in individuals’ post disease resolution, and remained least affected [116], suggesting a long-term persistence of IgA-mediated mucosal immunity. Furthermore, there have been reports of the generation of a multitude of nAbs, each targeting a specific domain of structural S protein. For instance, nAbs targeting RBD potentially interfere with ACE2 binding [117,118,119,120], as well as disrupt pre-fusion conformation [121], thereby thwarting cellular access by the virus and preventing infection and consequent disease.

Owing to the abovementioned facts, interest in the S protein regarding the development of vaccines, therapeutic mAbs, and nanobodies has been greatly enhanced, resulting in a trial of COVID-19 antibodies cocktails that may prove to be a better therapeutic approach [122]. To date, most of the potential nAbs generated in infected persons across the world have categorically been found to be specific for RBD; however, the NTD (N-terminal domain) of SARS-CoV-2 S protein also shows immunogenicity, leading to the generation of multitudes of antibodies due to its multiepitopic nature [123,124]. As per a recent study, out of 377 mAbs isolated from a sizeable cohort of SARS-CoV-2-infected patients, 80 showed a differential binding affinity towards RBD over five widely distributed clusters of epitopes. In addition, most potent antibodies also showed prophylactic and therapeutic responses in an animal model [124]. Together, such evidences indicate that both domains, i.e., RBD and NTD of the SARS-CoV-2 S protein, are highly immunogenic (albeit RBD is relatively more immunogenic) in nature, possessing multiple immunodominant epitopes and, therefore, both may be used for the development of a potential vaccine and/or raising therapeutic mAbs (monoclonal antibodies). In some recent studies, mAb isolated from recovered patients (named 4A8) showed a binding affinity for NTD (but not RBD), and was surprisingly found to be capable of neutralizing both authentic and pseudotyped SARS-CoV-2, possibly by imposing molecular constraints over conformational transitioning of S protein, indicating the immense possibility of NTD as another promising therapeutic target [123,124,125,126]. Moreover, nAbs (neutralizing antibodies) are also generated against the S2 subunit in infected individuals, indicative of it also being immunogenic in nature as well, [123] and may be crucial in vaccine design and development.

On the whole, a careful consideration-led compilation of multiple published scientific studies projected the percentage of seroconversion, i.e., the induction of virus-specific antibodies in infected persons in a range of 91–99%, with a 6+ months stability of anti-spike IgG and spike protein-specific memory B cells [127]. Surprisingly, Chi et al. concluded that there is not much positive correlation between the binding affinities of mAbs against RBD and virus neutralization; rather, mAbs against non-RBD epitopes, such as NTD and S2 subunits, also need to be taken into consideration while designing a therapeutic antibody cocktail to potentiate its overall efficacy [123]. In addition, mAbs isolated from convalescent patients showed a binding affinity towards homotrimeric S protein only [124], indicative of the formation of novel epitope(s) as a result of the molecular association of identical polypeptide subunits. Furthermore, the level of anti-S-expressing memory B cells and neutralization antibody titer showed a strong correlation with disease severity [128].

The S protein sequence of SARS-CoV-2 is 76% similar with that of SARS-CoV-1, whereas the NTD and RBD located within the S1 subunit, only showed a 50% and 74% similarity, respectively. Strikingly, there was a 90% protein sequence similarity with respect to the S2 subunit between these two CoVs (coronaviruses) [129]. The lack of ~100% conservation/or identity among prominent immunogenic domains, including NTD and RBD of the S protein of SARS-CoV-1 and 2, may be the underlying reason for the observed limited antibody cross-reactivity. In addition, the SARS-CoV-2 S protein conservation with MERS-CoV and seasonal HCoVs (human coronaviruses) is even lower, merely in a range of 19–21% [121], indicating a negligible, if any, likelihood of antibody cross-reactivity and, hence, almost no immune protection against the current SARS-CoV-2 following infections by these viruses, and eventual recovery. Apart from the B cell response and consequent antibody production, SARS-CoV-2 infection also leads to the induction of CD4+ and CD8+ T-cell responses, producing an array of cytokines and chemokines [127].

Contrary to negligible S protein-specific antibody-based cross-reactivity, there is a likelihood of T cell cross-reactivity, as a minor proportion of healthy individuals; not previously exposed to SARS-CoV-2, have been found to possess SARS-CoV-2-specific reactive T cells, owing to a past infection with ‘common cold’ coronaviruses [130]. Generally, it takes around two weeks post the onset of symptom for SARS-CoV-2-specific CD4^+^ and CD8^+^T cells to appear in peripheral blood circulation. Functionally, SARS-CoV-2-specific CD4^+^ T cells possess a central memory phenotype, whereas CD8^+^T cells are of a more effector type. Furthermore, T helper (T_H_) cells are prolific cytokine producer, and differentiate into two subtypes—T_H_1 and T_H_2—depending on the type of cytokines they produce. T_H_1 produces proinflammatory cytokines (IFN-λ, TNF-α, IL-2) with favourable antiviral properties, and are responsible for neutralizing intracellular parasites, whereas T_H_2 induces a humoral response by B cells, as well as secretes more of an anti-inflammatory response generating cytokines (IL-4, -5, -10, -13), thereby immunologically balancing and modulating the T_H_1 response [131]. Patients with SARS-CoV-2-induced ARDS often tend to have a T_H_1:T_H_2 ratio weighted towards the T_H_2 type, leading to substantial lung tissue damage [132,133]. Therefore, the ratio of T_H_1 to T_H_2 may be used as a diagnostic and prognostic marker during COVID-19, and a measure to balance the T_H_1 and T_H_2 responses may be a better therapeutic approach. Shedding light on underlying pathways leading to the induction of such humoral and cellular responses and understanding the associated foundational knowledge are very crucial steps to develop insights into virus pathogenesis, immunity, and recovery. Furthermore, it may also play an important role in design and development of vaccines. Similar to B cells, the development of virus-specific memory T-cells against the immunodominant S protein epitope, with a relatively long-term immunoprotective potential following a lethal infection of SARS-CoV-1, has been well established [134]. SARS-CoV-2-specific CD4+ (T_H_) and CD8+ (T_C_) T cells have also been detected in 70% and 100% of COVID-19-recovered individuals, respectively. Furthermore, CD4+ T cell responses to the S protein immunodominant epitope showed a positive correlation with the level of anti-SARS-CoV-2 IgG and IgA. The activation of the T-cell response may help in the clearance of viruses [135]. Of the total SARS-CoV-2-responsive CD4+ T cells, nearly 11–27% are accounted for the structural S protein as an antigen.

Dissimilar to the spike protein, other SARS-CoV-2 structural proteins, such as M (membrane) and E (envelope), show relatively less experimental immunogenicity with respect to their corresponding humoral responses. This may be attributed to their smaller molecular sizes, as well as a lesser protrusion of their ectodomains that are involved in recognition by immune cells [136]. Further, the N (nucleocapsid) protein is also likely to fail to evoke humoral response owing to its location inside the lipid bilayer envelope. Such findings are substantiated by the empirical observations, wherein mice (BALB/c and C57BL/6) immunized with Venezuelan equine encephalitis replicon particles (VRPs) expressing the SARS-CoV-2 S protein (VRP-S) showed a 1000 times reduction in SARS-CoV-2 titres by day 1 post infection (d.p.i) (3 weeks post vaccination). Protection against SARS-CoV-2 in immunized/vaccinated mice was found to be primarily provided by anti-S antibodies (most probably by blocking the attachment), as their sera transfer led to the accelerated clearance of the virus in naïve mice. In contrast, the immunization of mice with Venezuelan equine encephalitis replicon particles (VRPs) expressing membrane (VRP-M), nucleocapsid (VRP-N) and envelope (VRP-E) proteins did not yield similar results; also, their corresponding sera did not lead to the speedy clearance of the virus following SARS-CoV-2 infection [137]. Therefore, in light of the above empirical evidences, an ongoing vaccine development primarily focusses on the SARS-CoV-2 S protein rather than the M, N, and E proteins. However, there is a relatively higher protein sequence similarity of M or E proteins of SARS-CoV-2 with that of SARS-CoV-1 and MERS compared with RBD and the S protein. This suggests the possibility of a cross-reactivity-based immune protection against the current pandemic at both T_H_ and T_C_ levels, owing to the presence of T cell epitopes in former proteins (M and E), which were identified and published in previous studies [138]. Therefore, it is important and quite prudent to look into the possibility of the inclusion of M and E, along with the S proteins while designing the vaccine, as such an effort might be able to potentiate the existing vaccines’ efficacy, and provide holistic protection to immunized individuals against SARS-CoV-2 infection. Similar to the S protein, the most abundant viral N protein can also evoke both humoral and cellular immune responses owing to the presence of B cell and T cell epitopes [114,139]. For instance, Venezuelan equine encephalitis virus replicon particles (VRP) expressing N-specific CD4+ T cell epitope were found to provide complete protection against SARS-CoV-1 infection [140]. In addition to nAbs, a similar percentage of M and N protein-responsive CD4+ T cells have also been detected in recovered individuals. Further, reactive CD4+ T cells against non-structural proteins (NSPs), including NSP3, NSP4, ORF3a, and ORF8, have also been found. Eventually, SARS-CoV-2-specific CD4+ (T_H_) and CD8+ (Tc) cells, present in peripheral blood circulation, undergo a rapid decline (half-life = 3–5 months) owing to infiltration into lung tissue and virus-induced lysis [141], suggesting the non-permanence of the T cell-mediated immunity following natural infection and/or vaccination [127]. Nevertheless, corresponding memory T cells could live longer in secondary lymphoid organs to provide long-term immunity, and may protect the host during second and subsequent infections. Considering the above facts pertaining to immune responses, it may be prudent to design multi-epitope and multi-antigen-based vaccines, as well as the administration of multiple booster doses to combat the ongoing pandemic.

## 4. Protective Immune Response (Correlates of Protection) against SARS-CoV-2 for Vaccine Development

In general, a long-term immune-protection against any categories of pathogen requires the activation of both humoral and cellular immune responses, along with the formation of durable memory B and T cells. Therefore, it is important to develop understanding about correlates of protection (CoP) evoked against SARS-CoV-2 following infection, so as to decipher and leverage the operational underlying mechanism for therapeutic purposes, including vaccine design and development. CoP refers to the protective immune response requisite for the statistically interrelated protection of the host and, therefore, it may act as a predictor of a useful clinical outcome following natural infection and vaccination [142]. Hence, it may be employed as a rapid and an effective molecular tool to assess the protective immune response for a multitude of novel vaccine candidates against SARS-CoV-2 that are being developed worldwide. However, it is important to consider a well-established fact, that CoP evoked following a natural infection vis à vis vaccination differs with respect to certain immunological variables, both qualitatively and quantitatively, and this may be true in the case of COVID-19 as well, necessitating further empirical studies. A review of previously published work shows that CoP may be generated through either the humoral, cellular, or combined immune responses. Here, we tabulate the induction of humoral and cell-mediated CoP, following the administrations of a multitude of SARS-CoV-2 vaccines, as well as their currently known effectiveness, if any, against the globally reported SARS-CoV-2 variants (Table 3).

## 5. Vaccination, Herd Immunity (Population Immunity) and Herd Immunity Threshold

Developing a holistic understanding, regarding an intricate interrelation among vaccination, herd immunity, and the herd immunity threshold (HIT), is very crucial to contain the ongoing SARS-CoV-2 pandemic. Owing to the absence of effective and specific drugs against COVID-19, the immediate attention of researchers and clinicians is focussed on effective vaccine development, and the successful implementation of a vaccination programme to achieve herd immunity, also known as population immunity. Generally, an individual may acquire immunity either via natural infection or through vaccination. The herd immunity refers to the indirect protection of susceptible individuals of the population from an infectious disease when a certain percentage of the same population has gained immunity either through vaccination and/or natural infection. The well-studied, reliable, and quick method of establishing herd immunity to contain the spread of infectious disease, as well as provide indirect protection to susceptible/immunocompromised individuals, is through the rapid vaccination of individuals up to a herd immunity threshold level. This HIT level is an estimated upper limit, in terms of the percentage of a particular population, to the level of vaccination required to protect susceptible (yet unvaccinated) individuals, and contains the prevailing infection [155,156] (Figure 4).

The HIT for any infectious disease is often calculated mathematically using a formula, 1-1/R0, wherein R0 is referred to as the basic reproduction number. The R0 value is defined as the occurrence of the average number of secondary infections per infectious individual into a theoretically 100% susceptible (naïve or previously unexposed) population [155,156]. For instance, if we considered a hypothetical infectious pathogen with an R0 value of three, it would mean that there would be three secondary infections by each infected individual during the period of disease spread in a completely susceptible (naive) population (Figure 4A). The calculated herd immunity threshold in this case would be around 0.66 or 66%, meaning around 66% of individuals of this hypothetical population need to be vaccinated in order to contain and prevent the virus spread.

From the abovementioned formula, we could also derive a theoretical interpretation that the more infectious a pathogen, the greater is the associated R0, meaning higher upper limit to the level of vaccination to break the chain of transmission (Figure 4B). For instance, measles caused by a virus in the paramyxovirus family is extremely infectious with an R0 ranging between 12 and 18, requiring the vaccination for around 92–94% of the population [157]. For polio (clinically called as poliomyelitis), caused by the poliovirus, the threshold stands around 80%. Moreover, the maintenance and continuance of an infection require the R0 to be higher than unity. The R0 itself depends on the multiple variables [155]. Furthermore, there is a caveat associated with R0 as it presumes that every individual in a population is susceptible to the virus, which may be true at the outset of the disease outbreak, but with the progression of the pandemic/epidemic some people acquire immunity as a consequence of natural infection and eventual recovery. Therefore, we need to factor in this aspect as well, which is achieved by a related mathematical expression, called R effective (Re or Rt). In order to do so, one needs to apply the exponential growth method [158] by factoring in laboratory data with regard to the daily number of new cases (e.g., COVID-19), along with the most recent estimate of the serial interval (mean, standard deviation, and *p* = 0.05). Once these variables are known, one could use mathematical software R to obtain Re or Rt values [159]. Eventually, one could use the mathematical expression, Pcrit = 1 − (1/Rt), to arrive at a minimum (critical) percentage level of population immunity (both vaccine-and infection-acquired) required to stop the disease spread. Generally, at the outset of the disease outbreak, both R0 and Rt may remain close to each other; however, with the progress of infection, the Rt value declines. In addition, the values of both the R0 and Rt of SARS-CoV-2 are not yet precisely know; therefore, deciding the HIT with respect to COVID-19 is still awaited which, similar to other cases of other infectious pathogens, also shows geographical variation [160].

The accomplishment of herd immunity is based on the successful implementation of vaccination to the HIT level. However, we cannot afford to avoid factoring in immunogenicity and vaccine efficiency, as well as individual-specific immune response towards a vaccine. It is a well-known empirical fact that vaccination does not lead to proportionate immunization, and the reasons may entail vaccine efficiency and the nature of the working of the immune system of an individual. So far, none of the vaccines developed for COVID-19 has been found to be 100% efficient in terms of eliciting immune responses (humoral and cell-mediated); rather, most of them show an efficiency ranging from 50% to 95% against wild-type SARS-CoV-2 (Ref. No. NC_045512) (Table 1). This situation is further compounded due to the emergence of variant of concern (VOC), against which the efficiency of an existing vaccine is even further reduced drastically; some showing an ambiguous effect, whereas other stand negligibly effective as reported in preliminary clinical studies. For instance, Comirnaty (BNT162b2) and ChAdOx1 nCoV-19, two globally well-known and highly accepted vaccines, showed a 93.7% (95% CI, 91.6 to 95.3) and 74.5% (95% CI, 68.4 to 79.4) effectiveness, respectively, among persons with the alpha variant after the scheduled receipt of both doses of each vaccine. Similarly, Comirnaty (BNT162b2) and ChAdOx1 nCoV-19 showed 88.0% (95% CI, 85.3 to 90.1) and 67.0% (95% CI, 61.3 to 71.8) effectiveness of two doses, respectively, among those with the delta variant. In addition, the effectiveness of a single dose of these two vaccines against alpha and delta variants was quite similar, and stood only at around 48.7% (95% CI, 45.5 to 51.7) and 30.7% (95% CI, 25.2 to 35.7), respectively [161]. So far, the effectiveness of the Moderna COVID-19 vaccine (mRNA-1273) against SARS-CoV-2 VOC remains to be conclusively determined, and a study regarding the same is being carried out. Considering the abovementioned body of empirical evidences, achieving effective herd immunity would require further deliberation and insight development, incorporating ever emerging and evolving variants of concern (VOC) and variant of interest (VOI) (https://www.who.int/en/activities/tracking-SARS-CoV-2-variants/) (accessed on 17 October 2021).

## 6. Conclusions

The outbreak of SARS-CoV-2 in late 2019, Wuhan, the capital of China’s Hubei province, has attained a pandemic proportion, crippling socio-economic and health infrastructures, and causing a horrendous loss of lives, livelihoods, and academic loss worldwide. As of 23 September 2021, the three most affected nations in terms of reported cumulative infections, morbidity, and mortality are the USA, India, and Brazil. Such a horrific situation, as a consequence of a COVID-19 surge, has brought the entire world to a standstill, causing considerable mental health problems and a looming uncertainty of survival. Therefore, containing infections and reducing mortality occupy the utmost priority for various stakeholders, including policy makers, clinical researchers, doctors, paramedical staff, and the common people. This has led to a hunt for various novel drugs, repurposing multiple non-COVID-19 FDA-approved drugs intended for non-related infections, and design and development of prophylactic and/or therapeutic vaccines and cocktail of mAbs. While the search for therapeutic drugs is underway, the researchers have most reliably pinned their hopes on vaccines, which have even shown promising results in various phases of mandatory pre-clinical and clinical trials. Currently, around 20 vaccine candidates of multiple natures have received approval/authorization of the National (and some International) Regulatory Agencies (NRA), and are being used for vaccination worldwide. In addition, around 100 vaccines are under development and/or various phases of clinical trials; and, depending on trial outcomes, they are also likely to receive authorization in near future. The crucial aim of vaccines, irrespective of their intrinsic biological nature and ingredients (Figure 1), is to provide protection to individuals by evoking specific immune responses against SARS-CoV-2. Generally, both the innate and the adaptive immune responses are evoked following the entry of pathogens. As a consequence, an antiviral response is activated, which either could turn out to be protective or pathogenic depending upon multiple factors. Further, a large scale vaccination programme may play a crucial role in achieving the herd immunity threshold, which will protect even unvaccinated/susceptible and immunocompromised individuals via indirect immune protection. Lastly, it is increasingly believed that the likelihood of returning to pre-pandemic normalcy will greatly depend on the successful implementation of a global vaccination programme.

## Figures and Tables

**Figure 1 cells-10-02949-f001:**
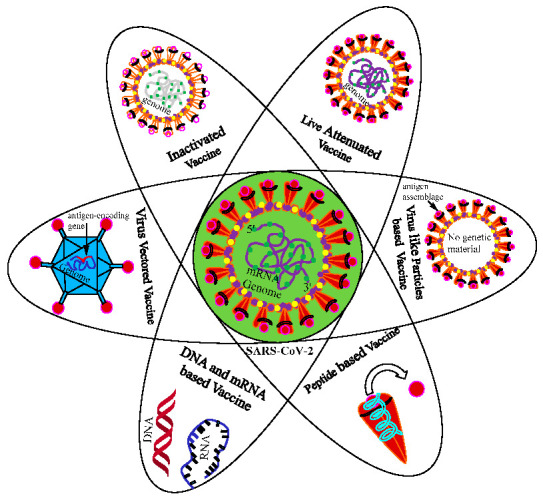
Multiple types of vaccine available and/or under various phases of development for COVID-19.

**Figure 2 cells-10-02949-f002:**
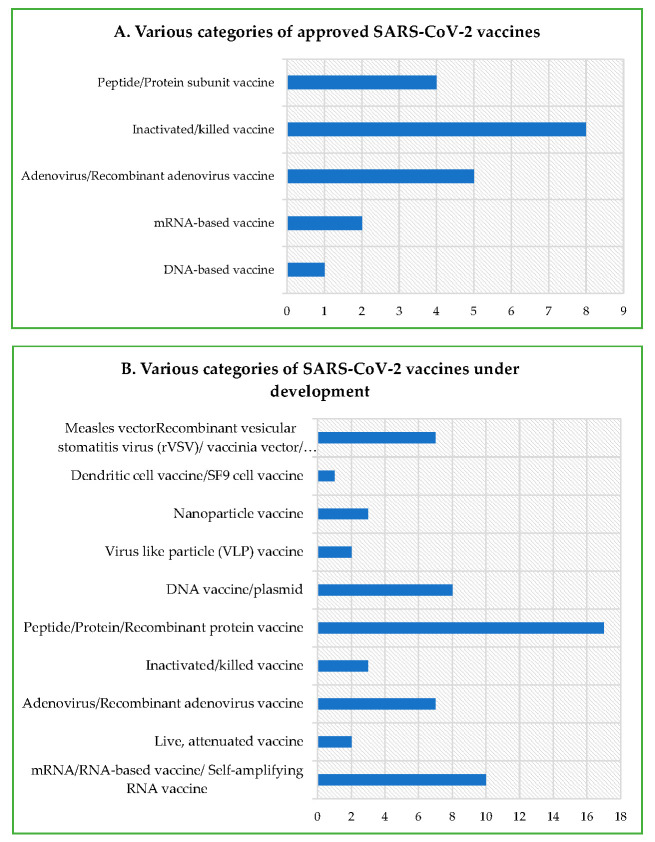
Nature of SARS-CoV-2/COVID-19 vaccines being developed and administered worldwide. (**A**) Nature/categories of authorized vaccines. (**B**) Nature of vaccines under various phases of pre-clinical and clinical development.

**Figure 3 cells-10-02949-f003:**
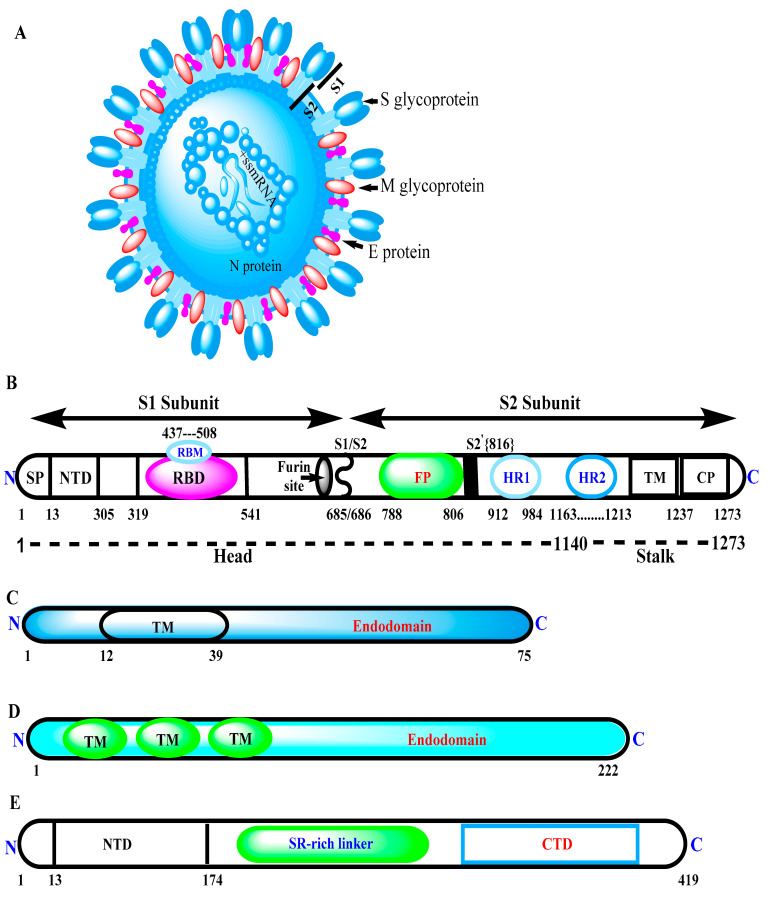
SARS-CoV-2 virion with its multiple structural proteins, namely, S, M, E, and N. (**A**) Structure of enveloped SARS-CoV-2 virion, depicting location of different structural proteins and +ss mRNA genome. (**B**) Structure of spike (S) protein, containing multiple domains and motifs in specific order, such as: N-terminus; SP—signal peptide; NTD—N-terminal domain; RBM—receptor-binding motif; RBD—receptor-binding domain; FP—fusion peptide; HR1—heptad repeat 1; HR2—heptad repeat 2; TM—transmembrane domain; CP—cytoplasm domain C-terminus. (**C**) Structure of envelope (**E**) protein, consisting of: N terminus; TM—transmembrane domain C-terminus. (**D**) Structure of membrane (M) protein, possessing N-terminus transmembrane domains TM and endodomain C-terminus. (**E**) Structure of N protein, consisting of: N-terminus; NTD-N—terminal domain; serine (SR)-rich linker region; NTD—N terminal domain; C-terminus. The position of each domain on S, M, E, and N may not be exactly scaled to amino acid residues owing to variability in published literatures. Even number of domains, as well as domain-specific function may be diverse.

**Figure 4 cells-10-02949-f004:**
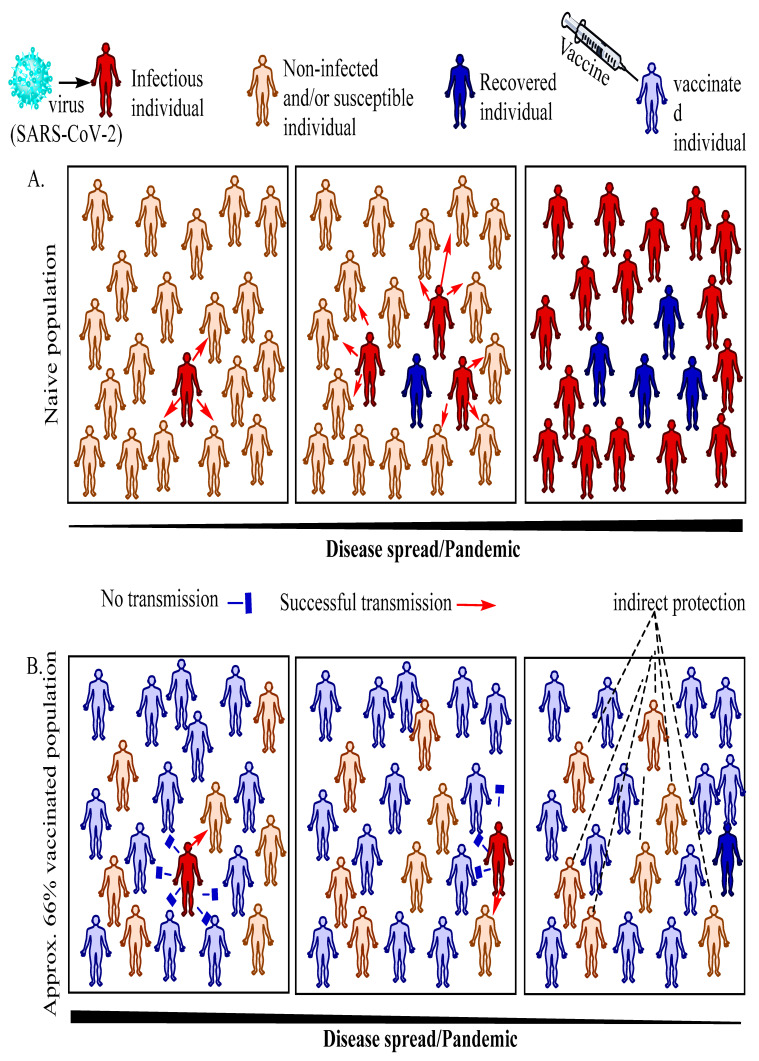
Schematic representation of dynamics of infectious diseases. Following introduction of one infected person in naïve population (hypothetical) consisting of majority of susceptible individuals (**A**) versus population consisting of 66% vaccinated individuals (66% is approximation and hypothetical value, which is actually decided considering the R0 that differs infectious disease-wise)/herd immunity (**B**). Following disease outbreak, the risk of contagion is very high, spreading rapidly through naïve population, whereas it is minimal in vaccinated population, protecting even susceptible individuals due to immune shield/barrier around them in the form of vaccinated individuals. Therefore, vaccination results into failure of virus spread and persistence in the population.

**Table 1 cells-10-02949-t001:** Important vaccines authorized/under development for COVID-19.

Brand Name	Current Dose/Gap and Route of Administration	Primary Developer(s)	Country/NRA	Clinical Trial Phase and Identifier	Approved/ Under Development	Reported Efficacy	Ref.
A. Inactivated or killed virus (SARS-CoV-2) vaccine (produced in Vero cells)							
CoronaVac (formerly PiCoVacc)	Two doses, between 14 and 18 days apart, intramuscular	Sinovac Biotech	China/NMPA	1 (NCT04352608) 2 (NCT04383574) 3 (NCT04456595)	Approved	Phase 3; 65.9%	[32]
BBIBP-CorV	Two doses, intramuscular injection	Sinopharm, Beijing Institute of Biological Products Co. Ltd.	China/NMPA	1 Not found 2 (ChiCTR2000032459) 3 (ChiCTR2000034780)	Approved	Phase 3; 86%	[33]
WIBP-CorV	Two doses, intramuscular injection	Wuhan Institute of Biological Products; China National Pharmaceutical Group (Sinopharm)	China/NMPA	1/2 (ChiCTR2000031809) Phase 3 trial is awaited	Approved	Phase 1/2; 72.5%	[30]
Covaxin (BBV152)	Two doses, 14 days apart, intramuscular	Bharat Biotech, Indian Council of Medical Research (ICMR), National Institute of Virology (NIV)	India/DCGI	1/2 (NCT04471519) Phase 3 trial is underway	Approved	Interim phase 3; 78%	[34]
CoviVac	Not specified	Chumakov Federal Scientific Center for the Research and Development of Immune and Biological Products of the Russian Academy of Sciences	Russia/Russian NRA	Phase 1/2 trial is underway	Approved	Not yet reported	Not yet
QazVac (QazCovid-in)	Two doses, 21 days apart, intramuscular	Research Institute for Biological Safety Problems	Kazakhstan	1/2 (NCT04530357) Phase 3 trial is underway	Approved	96%	Not yet
B. Live-attenuated vaccine against SARS-CoV-2							
Bacillus Calmette-Guerin (BCG) vaccine	Single dose, intradermally	University of Melbourne and Murdoch Children’s Research Institute; Radboud University Medical Center; Faustman Lab at Massachusetts General Hospital	Multinational	1 (NCT04328441) 2/3 (NCT04327206)	Not yet approved; under development	Not yet known	[35]
C. Adenovirus vector-based recombinant vaccine ** (Recombinant ChAdOx1 adenoviral vector encoding the SARS-CoV-2 spike protein antigen)* # (Human recombinant Adenovirus Vector (rAd5-S or rAd26-S) *encoding the SARS-CoV-2 spike protein antigen*) $ (Recombinant, replication-incompetent adenovirus type 26 (rAd26) vectored vaccine *encoding the SARS*-CoV-2 spike protein antigen) @ (Human recombinant Adenovirus Vector (rAd5-S) encoding the SARS-CoV-2 spike protein antigen)							
* COVID-19 Vaccine AstraZeneca/ (AZD1222) Vaxzevria/ Covishield	Two doses, between 4 and 12 weeks apart, intramuscular injection	AstraZeneca, University of Oxford, Serum Institute of India	United Kingdom (UK)/ EMA	1/2 (NCT04324606) 2/3 (NCT04400838) 3 NCT04516746	Approved	79% efficacy in phase 3 clinical trial (NCT04516746); 100% efficacy in severe disease and hospitalization patients	[36,37]
^#^ Sputnik V (formerly Gam-COVID-Vac Lyo) (rAd5-S or rAd26-S)	Two doses, 21 days apart, intramuscular injection	Gamaleya Research Institute, Acellena Contract Drug Research and Development	Russia/Russian NRA	1/2 (NCT04436471) and (NCT04436471) 3 (NCT04530396)	Approved	91.6% efficacy in phase 3 clinical trial	[38,39]
^#^ Sputnik light vaccine (rAd26-S)	No. of doses and gap are not yet finalized, intramuscular injection	Gamaleya Research Institute, Acellena Contract Drug Research and Development	Russia/Russian NRA	1/2 (NCT04713488) 3 (NCT04741061)	Approved	79.4% efficacy in phase 3 clinical trial	Not yet
^$^ COVID-19 Vaccine Janssen (JNJ-78436735; Ad26.COV2.S)	Single dose vaccine, intramuscular injection	Janssen vaccines (Johnsons & Johnsons)	The Netherlands, US/EMA	1/2 (NCT04436276) 3 (NCT04505722)	Approved	85% efficacy in phase 3 ENSEMBLE trial	[40,41]
^@^ Convidicea (Ad5-nCoV)	Single dose vaccine, but also evaluated in trial with 2 doses, intramuscular	CanSino Biologics	China/EMPA	1 (NCT04313127) 2 (NCT04341389) 3 (NCT04526990)	Approved	65.7% efficiency in interim phase 3 clinical trial	[42]
D. *mRNA vaccine* (BNT162b2 is a lipid nanoparticle–formulated, nucleoside-modified mRNA vaccine encodes prefusion spike protein) (mRNA-1273 encodes the prefusion-stabilized S protein of SARS-CoV-2) (ARCoV: lipid nanoparticle-encapsulated mRNA (mRNA-LNP) encodes the receptor-binding domain (RBD) of SARS-CoV-2)							
Comirnaty (formerly BNT162b2)	Two doses, 21 days apart, intramuscular injection	Pfizer, BioNTech; Fosun Pharma	Multinational/EMA	1/2 (NCT04380701) 2 (NCT04649021) 2/3 (NCT04368728)	Approved	~90% efficacy in phase 3 clinical trail	[43,44]
Moderna COVID-19 Vaccine (mRNA-1273)	Two doses, 28 days apart, intramuscular injection	Moderna, BARDA, NIAID	The USA/EMA	1 (NCT04283461) 2 (NCT04405076) 3 (NCT04470427)	Approved	~94.1% efficacy in phase 3 clinical trial	[45,46]
ARCoV	Intramuscular injection	Academy of Military Medical Sciences, Walvax Biotechnology, Suzhou Abogen Biosciences	China/NMPA	ChiCTR2000034112	Under development	Not yet reported	[47]
E. *peptide/subunit Vaccine*							
EpiVacCorona	Two doses, 21–28 days apart, intramuscular injection	Federal Budgetary Research Institution State Research Center of Virology and Biotechnology	Russia/Russian NRA	1/2 (NCT04527575) 3 (NCT04780035)	Approved	Not yet reported	Not yet
SCB-2019(stabilized trimeric form of the spike (S)-protein (S-Trimer)	Two doses, 21 days apart, intramuscular	Glaxo SmithKline, Sanofi, Clover Biopharmaceuticals, Dynavax and Xiamen Innovax	Australia	1 (NCT04405908) 2/3 is underway	Under development	Not yet reported	[48]
F. *DNA Vaccine* (Plasmid DNA expressing S protein*)*							
INO-4800	Two doses, intradermal injection	INOVIO Pharmaceuticals, International Vaccine Institute	USA	1 (NCT04336410) 2/3 (NCT04642638)	Under development	Not yet specified	[49]
AG0301-COVID-19	Two doses, 14 days apart, intramuscular injection	AnGes, Inc.	Japan	1/2 (NCT04463472)	Under development	Not yet specified	Not yet
GX-19N	Two doses, 29 days apart, intramuscular injection	Genexine	South Korea	1/2a (NCT04715997)	Under development	Not yet specified	Not yet
CORVax12	Two doses, 28 days apart, DNA electroporation	OncoSec; Providence Cancer Institute	The USA	1 (NCT04627675)	Under development	Not yet specified	Not yet
G. *Virus-like particle (VLP) or nanoparticle vaccine*							
ABNCoV2	Two doses, 28 days apart, intramuscular injection	ExpreS2ion Biotech; Bavarian Nordic A/S	Netherlands	1 (NCT04839146)	Under development	Not yet specified	Not yet
SpFN (spike ferritin nanoparticle vaccine)	Doses and gap are unspecified, intramuscular injection	US Army Medical Research and Development Command	The USA	1 (NCT04784767)	Under development	Not yet specified	Not yet

NRA: National Regulatory Authority; NMPA: National Medical Products Administration; DCGI: Drugs Controller General of India; EMA: European Medicines Agency. All the data in above table have been accessed and updated upto 23 September 2021.

**Table 2 cells-10-02949-t002:** Predicted average antigenic propensity of SARS-CoV-2 structural proteins (S, M, E, and N). The sequences of structural proteins were retrieved individually as FASTA format from NCBI database that is curated and designated for SARS-CoV-2 (SARS-CoV-2 Resources—NCBI (nih.gov). The sequences were uploaded individually to antigenic propensity prediction tool (http://imed.med.ucm.es/Tools/antigenic.pl, accessed on 17 October 2021) for calculation of their respective average antigenicity score. The score is considered as crucial information while designing and developing myriad of vaccines.

Name of Structural Proteins	Length (Amino Acids)	Predicted Average AntiGenic Propensity Score	NCBI Ref. Sequence
Spike (S) glycoprotein	1273	1.0146	YP_009724390.1
Membrane (M) glycoprotein	222	1.0532	YP_009724393.1
Envelope (E) protein	75	1.1202	YP_009724392.1
Nucleocapsid (N) phosphoprotein	419	0.9871	YP_009724397.2

**Table 3 cells-10-02949-t003:** Important approved vaccines and reported immune responses.

Vaccine	Humoral Response (IgG) (Wild-type SARS-CoV-2)	Cellular Response (Wild type-SARS-CoV-2)	Reported Effectiveness against SARS-CoV-2 Variants of Concern (VOC)	Ref.
CoronaVac (formerly PiCoVacc)	Induction of specific IgG against S and N proteins, RBD in mice, rats, and non-human primates (pre-clinical); induction of anti-RBD IgG and nAbs in humans (Clinical)	No detectable induction of T cell response (T_H_1 or T_H_2) cell responses in NHPs as well as human	Effective against D614G, and B.1.1.7 Less effective against B.1.351	[29,143,144]
BBIBP-CorV	Induction of nAbs in mice, rats, rabbits, guinea pigs, NHPs (Macaca fascicularis and Rhesus macaques), and humans	No induction of either T_H_1 or T_H_2 cell responses in NHPs	Effective against B.1.1.7 Less effective against B.1.351	[33,34]
WIBP-CorV	Formation of virus-specific IgG and nAbs in humans	No report of specific induction of either T_H_1 or T_H_2 cell responses in NHPs	Not yet known/reported	[30]
Covaxin (BBV152)	Neutralizing antibody (nAbs) response in humans	T cell responses, with biasness towards T_H_1 cells	Effective against B.1.1.7; effective against B.1.617	[34]
COVID-19 Vaccine AstraZeneca/ (AZD1222) Vaxzevria/ Covishield	Induction of anti-S antibody and nAbs in mice, NHPs, as well as humans, with nAb titres similar to convalescent plasma	Induction of high T_H_1 cell, but low T_H_2 cell responses in mice	Reduced neutralisation activity against the B.1.1.7 variant in vitro; however, effective against B.1.1.7 in vivo	[145,146,147]
Sputnik V (formerly Gam-COVID-Vac Lyo) (rAd5-S or rAd26-S)	Induction of both RBD-specific antibody and nAbs in humans	Induction of T_H_ and T_c_ cell responses	Significant neutralizing activity against B.1.1.7, B.1.351, P.1, B.1.617.2 and B.1.617.3	[39,148]
COVID-19 Vaccine Janssen (JNJ-78436735; Ad26.COV2.S)	Generation of both RBD-specific and neutralizing antibodies in hamsters and NHPs	Induction of high T_H_1, but low T_H_2 cell responses in NHPs	Effective against B.1.617.2	[40,41,149,150]
Convidicea (Ad5-nCoV)	Generation of RBD-specific and neutralizing antibodies in humans	Generation of T_H_1 cell response	Not yet known/reported	[42,151]
Comirnaty (formerly BNT162b2)	Generation of RBD-specific and neutralizing antibodies (nAbs) in humans	Not yet known	Effective against B.1.526, B.1.429 and B.1.1.7 variants	[43,152]
Moderna COVID-19 Vaccine (mRNA-1273)	Generation of S-specific and nAbs in mice, NHPs, and humans	Induction of high T_H_1, but low T_H_2 cell responses in mice, NHPs and human	Effective against B.1.351 and P.1 variants; this vaccine also neutralizes the B.1.617.1 variant, albeit 6.8-fold less effectively	[45,153,154]

Note: The humoral response following vaccine administration was quantified by measurement of virus and/or virus-related specific immunoglobulin G (IgG), whereas T_H_1 and T_H_2 cell responses were measured by detection of their respective cytokines, such as IFNγ, IL-2, and TNF (T_H_1); IL-4, IL-5, IL-6, IL-10, and IL-13 (T_H_2).

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
