# Peer review of "Insights into COVID-19 Vaccine Development Based on Immunogenic Structural Proteins of SARS-CoV-2, Host Immune Responses, and Herd Immunity"

_cells, 2021, doi:10.3390/cells10112949_

Round 1
Reviewer 1 Report
Dear authors,
The manuscript “Insights into COVID-19 vaccine development based on immunogenic structural proteins of SARS-CoV-2, host immune responses and herd immunity” is a very good review with detailed description of SARS-CoV-2 structure, and their action to induce immune responses. This manuscript will be very helpful for the scientific community. There are some few suggestions to improve this work and you can find my comments below.
English language needs some improvement. There are typos and misspelling words in all manuscript. An example is this kind of mistake “protcetive” instead of “protective”(line 132). Please check the main text.
Line 385-388: Authors mentioned that SARS-CoV-2 specific CD4+ and CD8+ T cells decrease rapidly after infection or vaccination. This is not true. The reality is based on the percentage of those specific cells going down from peripheral blood, but those cells profile, as memory T cells, are maintained in lymphoid organs. The T cells do not have the property to be activated forever in blood, they will die, to renew the repertoire, and they will be activated as soon as a new contact with the virus or a similar antigen again, as occur for all vaccines worldwide. I would rephrase and explain better this paragraph, because the reader could have the wrong idea that T cells disappear from our bodies, and this is not a real thing.
Line 389-474: I would not insert the adaptive immune response before innate immune response. Innate immune response is always the first line of cellular defense, and then, the antibodies and T cells. Maybe, it will be better to organize the immune events, as 1) innate, 2) humoral, and 3) adaptive immune responses. Some studies also have been describing that innate cell response is related to rapid recovery or to poor outcome, as severity in COVID-19. Given that, the innate immune responses are also considered a target for the development of COVID-19 treatments. Please, see these manuscripts, they might be helpful DOI: 10.1111/all.14364; DOI: 10.1002/eji.202048693; DOI: 10.1155/2020/8827670.
Table 2: It is important to note that in this table the authors are providing information about helper CD4+ T immune responses. I cannot find any explanation about the difference among Th1 and Th2 in the manuscript. It is very important to the readers to know that Th1 responses has benefits to the host regard to a pathogenicity of a viral infection, as Coronaviruses did. The inflammatory responses, and damage to the organs, as lungs, are related to Th2 immune responses. If you have vaccines able to activate Th1 response, but not Th2 response, this is extremely good. If you have a vaccine inducing Th2 response, this is bad. So, this information should be included in the manuscript, since the authors have a table describing which each vaccine is able to induce in cellular immune responses. Please, see these manuscripts, they might be helpful doi: 10.1155/2020/8827670; DOI: 10.1128/JVI.06048-11.
Author Response
Responses to 1st Reviewer’s Comments
Dear authors,
The manuscript “Insights into COVID-19 vaccine development based on immunogenic structural proteins of SARS-CoV-2, host immune responses and herd immunity” is a very good review with detailed description of SARS-CoV-2 structure, and their action to induce immune responses. This manuscript will be very helpful for the scientific community. There are some few suggestions to improve this work and you can find my comments below.
- English language needs some improvement. There are typos and misspelling words in all manuscript. An example is this kind of mistake “protcetive” instead of “protective”(line 132). Please check the main text.
Response: First of all, authors express their collective gratitude towards reviewer for invaluable suggestions and comments. Authors have revised the entire manuscript both grammatically and structurally to maintain the seamless flow of information, as well as updated the information wherever needed.
- Line 385-388: Authors mentioned that SARS-CoV-2 specific CD4+ and CD8+ T cells decrease rapidly after infection or vaccination. This is not true. The reality is based on the percentage of those specific cells going down from peripheral blood, but those cells profile, as memory T cells, are maintained in lymphoid organs. The T cells do not have the property to be activated forever in blood, they will die, to renew the repertoire, and they will be activated as soon as a new contact with the virus or a similar antigen again, as occur for all vaccines worldwide. I would rephrase and explain better this paragraph, because the reader could have the wrong idea that T cells disappear from our bodies, and this is not a real thing.
Response: The sentences have been reframed in the light of suggestion, making them clearer and more accurate.
- Line 389-474: I would not insert the adaptive immune response before innate immune response. Innate immune response is always the first line of cellular defense, and then, the antibodies and T cells. Maybe, it will be better to organize the immune events, as 1) innate, 2) humoral, and 3) adaptive immune responses. Some studies also have been describing that innate cell response is related to rapid recovery or to poor outcome, as severity in COVID-19. Given that, the innate immune responses are also considered a target for the development of COVID-19 treatments. Please, see these manuscripts, they might be helpful DOI: 10.1111/all.14364; DOI: 10.1002/eji.202048693; DOI: 10.1155/2020/8827670.
Response: Having considered reviewer’s appropriate suggestion, there has been restructuring regarding these two subsections, placing innate immune response first, followed by adaptive immune response. Besides, the role of innate immunity has been revised and updated factoring in above suggestions.
- Table 2: It is important to note that in this table the authors are providing information about helper CD4+ T immune responses. I cannot find any explanation about the difference among Th1 and Th2 in the manuscript. It is very important to the readers to know that Th1 responses has benefits to the host regard to a pathogenicity of a viral infection, as Coronaviruses did. The inflammatory responses, and damage to the organs, as lungs, are related to Th2 immune responses. If you have vaccines able to activate Th1 response, but not Th2 response, this is extremely good. If you have a vaccine inducing Th2 response, this is bad. So, this information should be included in the manuscript, since the authors have a table describing which each vaccine is able to induce in cellular immune responses. Please, see these manuscripts, they might be helpful doi: 10.1155/2020/8827670; DOI: 10.1128/JVI.06048-11.
Response: The roles of Th1 and Th2 cells have been explained in context of COVID-19.

Reviewer 2 Report
The manuscript by Chaudhary and colleagues described advances in design of COVID-19 vaccines and their development. It also discussed innate and adaptive immune responses to SARS-cov-2 infection, and the occurrence of herd immunity following natural infection or/and vaccination. To the reviewer, the manuscript is too generic, and the potential readers do not get into the deep of any of issues treated. Although it reads very well but it sounds like a section of a book and lack of deep discussions and original thinking. I appreciate the amount of information provided by the authors, but they certainly need to have their own comments and intellectual inputs incorporated into the manuscript. The manuscript should be revised, entirely, to avoid repeating textbook materials.
A few minors:
lines 197 to 214: Please discuss the potential problems associated with mRNA vaccines e.g. mRNA instability and critical impact of post-manufacturing logistics in vaccine efficacy. Additionally, review the potential risk of plasmid uptake and integration into host DNA following vaccination with a DNA vaccine.
As it is entitled “Insight into COVID-19 vaccine development …”, a clear discussion and conclusion on their design, safety, and finally efficacy, specifically against emerging variant of SARS-Cov-2 is expected.
Among all existing SARS-Cov-2 vaccines, which one has the highest chance to control the current pandemic? Any comments?
There is an extra space in line 334 and 501
Author Response
Response to 2nd Reviewer’s comments
The manuscript by Chaudhary and colleagues described advances in design of COVID-19 vaccines and their development. It also discussed innate and adaptive immune responses to SARS-cov-2 infection, and the occurrence of herd immunity following natural infection or/and vaccination. To the reviewer, the manuscript is too generic, and the potential readers do not get into the deep of any of issues treated. Although it reads very well but it sounds like a section of a book and lack of deep discussions and original thinking. I appreciate the amount of information provided by the authors, but they certainly need to have their own comments and intellectual inputs incorporated into the manuscript. The manuscript should be revised, entirely, to avoid repeating textbook materials.
Response: First and foremost, authors express their gratitude towards your time and effort for reviewing and providing invaluable suggestions with regard to the manuscript. Keeping comments in mind, there have been revision of the entire manuscript to improve, elaborate and update the overall content in the light of ever emerging empirical knowledge. Furthermore, authors have put forth their own critical comments as and when required.
A few minors:
- lines 197 to 214: Please discuss the potential problems associated with mRNA vaccines e.g. mRNA instability and critical impact of post-manufacturing logistics in vaccine efficacy. Additionally, review the potential risk of plasmid uptake and integration into host DNA following vaccination with a DNA vaccine.
Response: Potential risks associated with both plasmid DNA and mRNA vaccines have been discussed.
- As it is entitled “Insight into COVID-19 vaccine development …”, a clear discussion and conclusion on their design, safety, and finally efficacy, specifically against emerging variant of SARS-Cov-2 is expected.
Response: Suggestions have been followed and incorporated on the basis of available literature.
- Among all existing SARS-Cov-2 vaccines, which one has the highest chance to control the current pandemic? Any comments?
Response: Considering continuous emergence of multiple novel SARS-CoV-2 variants around the world, and non-availability of enough empirical evidences on effectiveness of currently available vaccine against SARS-CoV-2 variant of concern (VOC), it would be imprudent to answer above question with certainty. However, some studies have been carried out, showing differential and reduced effectiveness of currently available vaccines against different VOC (alpha, beta, gamma and delta). They also reported that effectiveness increases significantly after receipt of both doses of vaccine. For instance, Comiranty (BNT162b2) and ChAdOx1 nCoV-19 show 93.7% (95% CI, 91.6 to 95.3) and 74.5% (95% CI, 68.4 to 79.4), effectiveness, respectively, among persons with the alpha variant; and 88.0% (95% CI, 85.3 to 90.1) and 67.0% (95% CI, 61.3 to 71.8) effectivness, respectively, amongst those with the delta variant after receipt of both doses (N Engl J Med 2021; 385:585-594 DOI: 10.1056/NEJMoa2108891). No conclusive report on effectiveness of Moderna COVID-19 vaccine against variants is yet available (N Engl J Med 2021; 384:1468-1470 DOI: 10.1056/NEJMc2102179). Other approved vaccines are undergoing trials against SARS-CoV-2 VOC and conclusive results are awaited.
- There is an extra space in line 334 and 501
Response: The extra spaces, including the one pointed out, have been removed.

Reviewer 3 Report
- The Abstract does not cover the main topics reviewed in the manuscript. In addition to a summary of the vaccines approved and under development, the manuscript also discussed the adaptive and innate immunity evoked by the vaccines/virus, and, herd immunity. These should also be included in the Abstract.
- Some literature is not relative to the points in the main text. For example, Ref 41 in Table 1 is barely relevant to the information in the Table. Also, Ref4 does not support the statements made in Line 58-59.
Some literature was misinterpreted. For example, in Section 2.2 (Table 1), BCG is reported possibly to have some effects in defense against SARS-COV-2 (Table 1, Ref 44), but not a SARS-COV-2 vaccine under development.
These could cause damage to the credibility of the manuscript, especially for a review. Authors need to check the citations carefully.
- The authors categorized the vaccines into several classes. However, the category was not followed in Figure 2 B. It should be consistent with Table 1.
- Section 3 discussed the immunogenic properties and adaptive/ innate immunity response evoked by vaccine/virus. Thus, the current caption for this section, “The SARS-CoV-2 structural and non-structural proteins as targets against COVID-19”, cannot recapitulate the section.
- Line 239-247 described the tools to predict the antigenic propensity, therefore should not be put in the main text, instead, better put it in the table legend.
- When discussing herd immunity, the efficacy of the vaccines should be taken into account.
- The writing is generally clear, however, it is a bit lengthy. Expressions such as “conservation/or identity”, “repurposing/repositioning”, “proportion/percentage” etc., make the manuscript less readable.
- The text need careful editing to avoid errors such as “STAT279”, “IFN-7” etc. Two Tables have the same number, Table 2.
Author Response
Response to 3rd Reviewer’s comments
Comments and Suggestions for Authors
- The Abstract does not cover the main topics reviewed in the manuscript. In addition to a summary of the vaccines approved and under development, the manuscript also discussed the adaptive and innate immunity evoked by the vaccines/virus, and, herd immunity. These should also be included in the Abstract.
Response: First of all, authors would like to collectively express their gratitude towards reviewer for invaluable comments and suggestions.
Abstract has been updated as per suggestion to make it more inclusive and recapitulating.
- Some literature is not relative to the points in the main text. For example, Ref 41 in Table 1 is barely relevant to the information in the Table. Also, Ref4 does not support the statements made in Line 58-59.
Response: In light of latest knowledge, data with respect to Ref 41, showing 50% earlier efficiency has been updated with latest vaccine efficiency 65.9% as per result published on 2 September 2021 in NEJM (N Engl J Med 2021; 385:875-884
DOI: 10.1056/NEJMoa2107715). Further, statement has been corrected in accordance with Ref.4.
- Some literature was misinterpreted. For example, in Section 2.2 (Table 1), BCG is reported possibly to have some effects in defense against SARS-COV-2 (Table 1, Ref 44), but not a SARS-COV-2 vaccine under development. These could cause damage to the credibility of the manuscript, especially for a review. Authors need to check the citations carefully.
Response: The section 2.2 has been corrected and reinterpreted with proper citations. Authors highly appreciate reviewer for pointing mistakes, which would otherwise have been left misinterpreted.
- The authors categorized the vaccines into several classes. However, the category was not followed in Figure 2 B. It should be consistent with Table 1.
Response: A DNA category has been added to Fig.2 A as it received approval quite recently. This addition has made bar graph more representative of vaccine categories.
- Section 3 discussed the immunogenic properties and adaptive/ innate immunity response evoked by vaccine/virus. Thus, the current caption for this section, “The SARS-CoV-2 structural and non-structural proteins as targets against COVID-19”, cannot recapitulate the section.
Response: The earlier title “The SARS-CoV-2 structural and non-structural proteins as targets against COVID-19” has been replaced by more appropriate and recapitulating title “Immune responses to SARS-CoV-2 structural and non-structural proteins”.
- Line 239-247 described the tools to predict the antigenic propensity, therefore should not be put in the main text, instead, better put it in the table legend.
Response: Rearranged as per the reviewer’s suggestion
- When discussing herd immunity, the efficacy of the vaccines should be taken into account.
Response: Following reviewer’s relevant suggestion, the efficiency of some globally well accepted authorized vaccines against wild type virus and variant of concern has been discussed.
- The writing is generally clear, however, it is a bit lengthy. Expressions such as “conservation/or identity”, “repurposing/repositioning”, “proportion/percentage” etc., make the manuscript less readable.
Response: Authors have removed and rectified repetitive and synonymous words to make manuscript more readable.
- The text need careful editing to avoid errors such as “STAT279”, “IFN-7” etc. Two Tables have the same number, Table 2.
Response: Text has been revised carefully to avoid such typos and errors. Table numbering has been corrected.

Round 2
Reviewer 1 Report
The authors have been made the corrections and improvements as requested.
Author Response
Response to Reviewer’s comments (Second Round)
Reviewer’s comment: The authors have been made the corrections and improvements as requested.
Authors’ response: Authors collectively thank to the Reviewer for his/her invaluable comments and suggestions made with respect to our submitted manuscript. We would like to make a submission that following receipt of first round comments from reviewer 1, we revised our manuscript incorporating all the suggestions, and submitted the same on 24 September 2021. In response to our submitted revision, the reviewer has put forth abovementioned comment, suggesting work has been improved.
Reviewer 2 Report
The authors revised their manuscript to add more information, especially the parts that were missed in the previous version. Still, it is too generic.
Author Response
Response to Reviewer’s comments (Second Round)
Reviewer’s comment: The authors revised their manuscript to add more information, especially the parts that were missed in the previous version. Still, it is too generic.
Authors’ response: Authors collectively thank to the Reviewer for his/her invaluable comments and suggestions made with respect to our submitted manuscript. We would like to make a submission that following receipt of first round comments from reviewer 2, we revised our manuscript incorporating all the suggestions, and submitted the same as separate response on 24 September 2021. In response to our submitted revision, the reviewer has put forth abovementioned comment, suggesting work has been improved with addition of more information.